# Energy-selective confinement of fusion-born alpha particles during internal relaxations in a tokamak plasma

A. Bierwage [1,2 ✉], K. Shinohara [2,3], Ye.O. Kazakov [4], V. G. Kiptily[5,11], Ph. Lauber[6,11], M. Nocente [7,8], Ž. Štancar [5,9,11], S. Sumida[2], M. Yagi[1,12], J. Garcia [10,12], S. Ide[2,12] & JET Contributors*

Long-pulse operation of a self-sustained fusion reactor using toroidal magnetic containment requires control over the content of alpha particles produced by D-T fusion reactions. On the one hand, MeV-class alpha particles must stay confined to heat the plasma. On the other hand, decelerated helium ash must be expelled before diluting the fusion fuel. Here, we report results of kinetic-magnetohydrodynamic hybrid simulations of a large tokamak plasma that confirm the existence of a parameter window where such energy-selective confinement can be accomplished by exploiting internal relaxation events known as sawtooth crashes. The physical picture — a synergy between magnetic geometry, optimal crash duration and rapid particle motion — is completed by clarifying the role of magnetic drifts. Besides causing asymmetry between co- and counter-going particle populations, magnetic drifts determine the size of the confinement window by dictating where and how much reconnection occurs in particle orbit topology.

[1] National Institutes for Quantum Science and Technology (QST), Rokkasho Fusion Institute, Rokkasho, Aomori, Japan. [2] National Institutes for Quantum Science and Technology (QST), Naka Fusion Institute, Naka, Ibaraki, Japan. [3] Department of Complexity Science and Engineering, The University of Tokyo, Kashiwa, Chiba, Japan. [4] Laboratory for Plasma Physics, LPP-ERM/KMS, Partner in the Trilateral Euregio Cluster (TEC), Brussels, Belgium. [5] United Kingdom Atomic Energy Authority, CCFE, Culham Science Centre, Abingdon, United Kingdom. [6] Max-Planck-Institut für Plasmaphysik, Garching, Germany. [7] Dipartimento di Fisica 'G. Occhialini', Università di Milano-Bicocca, Milano, Italy. [8] Institute for Plasma Science and Technology, National Research Council, Milan, Italy. [9] Jožef Stefan Institute, Ljubljana, Slovenia. [10] CEA, IRFM, 13108 Saint-Paul-lez-Durance, France. [11] These authors contributed equally: V. G. Kiptily, Ph. Lauber, Ž. Štancar. [12] These authors jointly supervised this work: M. Yagi, J. Garcia, S. Ide. *A list of authors and their affiliations appears at the end of the paper. ✉email: bierwage.andreas@qst.go.jp

Laboratories around the world have intensified R&D activities for experimental reactors that should demonstrate the practical feasibility of extracting useful energy from controlled nuclear fusion. Spearheaded by ITER[1], the tokamak concept is the present mainstream approach to magnetically confined fusion (MCF). While their use as a power plant still awaits breakthroughs, the accumulated scientific evidence suggests that tokamaks can produce a burning plasma, where fusion reactions are self-sustained for times much longer than the confinement times of thermal energy and charged particles.

Tokamaks use a strong magnetic field to confine hydrogen isotope plasmas with high temperatures (~10 keV) in a toroidal volume as sketched in Fig. 1a. The helically wound $\mathbf{B}$ field consists of a dominant toroidal component $\mathbf{B}_{tor}$ that is provided by external coils and a weaker poloidal component $\mathbf{B}_{pol}$ that is induced by electric currents carried by the plasma itself. For a toroidal surface with long circumference $2\pi R$ and short circumference $2\pi\bar{r}$, the mean helical pitch of the magnetic field vector $\mathbf{B}$ is given by $q \approx \bar{r}B_{tor}/(RB_{pol})$, where $R$ is the major radius of the torus and $\bar{r}$ the mean minor radial distance from the center of the plasma. In preparation for a later generalization, we call $q$ the 'field helicity'[2].

The field helicity profile $q(\bar{r}) \propto 1/I_{tor}(\bar{r})$ varies across the plasma radius, in inverse proportion to the plasma's electric current profile $I_{tor}(\bar{r})$. Each toroidal surface where the field helicity has a rational value $q = m/n$ represents a geometric resonance. Resonances with small integers $m$ and $n$ can facilitate macroscopic long-lived plasma distortions and instabilities. Most notably, when the plasma current distribution reaches a certain threshold, so that the field helicity drops below unity ($q < 1$) somewhere in the plasma, a self-organization process sets in that prevents further steepening of the current density profile[3]. This results in a quasi-steady state, which future fusion reactor experiments like ITER[1] will exploit.

This self-organization process can be pictured as follows. $q = 1$ means that magnetic field lines close on themselves after one poloidal and one toroidal turn as illustrated in Fig. 1a. In our

example, which has the dimension of the Joint European Torus (JET), the condition $q < 1$ is satisfied within a radius of about $\bar{r} \approx 0.2$ m as indicated by the dashed lines in Fig. 1b, c. The portion of the plasma located within the $q = 1$ surface can be easily displaced by the destabilizing forces associated with gradients in the current density. The resulting perturbation, which is known as internal kink mode, has the form of a tilted torus within a torus. In other words, the kinked $q = 1$ torus is resonant with the toroidal geometry of the tokamak as a whole. Together with mechanisms facilitating magnetic reconnection[4], the formation of a region with $q < 1$ gives rise to quasi-periodic relaxation events that can be observed in the form of sawtooth oscillations in time traces of the central electron temperature $T_e$ as in Fig. 2. These data were acquired during a JET pulse, where the 3-ion radio-frequency (RF) heating scheme was applied to a mixed D-$^3$He plasma, successfully generating and confining fusion-born alpha particles[5–7].

Benign sawtooth activity is considered to have beneficial effects in fusion-oriented applications. Besides helping to keep the core plasma near a well-defined state, this mixing process was found to prevent excessive accumulation of heavy-ion impurities that would cause radiative cooling[8]. By the same token, it has repeatedly been proposed to use sawteeth for the expulsion of helium ash (henceforth called 'slow alphas') from the core of a deuterium-tritium (D-T) fusion reactor[9–11].

There is however a caveat: It is not desirable to have sawteeth that flatten the profiles of all particles. In particular, energetic $^4_2$He$^{2+}$ ions ('fast alphas') should ideally be left unperturbed, since they provide the heating power in a self-sustained burning fusion plasma. In the case of so-called alpha channeling via radio-frequency (RF) waves[12,13], it is envisioned that the alpha particles gradually diffuse outward while transferring their energy to thermal ions via damped plasma waves. A recent study shows, however, that the RF-cooled alphas may remain in the plasma core[14]. In contrast, resonant interactions with the internal kink and other low-frequency magnetohydrodynamic (MHD) modes can cause rapid ballistic transport before the alphas have transferred their energy to the bulk[15].

After decades of research — especially during the 1990s when the Tokamak Fusion Test Reactor (TFTR) and JET operated with D-T plasmas[16,17] — this conundrum of needing to confine fast alphas while expelling slowed-down helium ash is still being actively explored. A driving force behind these studies was the insight that sufficiently energetic charged particles can decouple from the dynamics of the bulk of a magnetized plasma, which has been known since the early years of MCF research (see ref. [18] for a review). The energy threshold above which ions decouple from the internal kink mode has been estimated theoretically by Kolesnichenko and Yakovenko[19] and confirmed experimentally by Muscatello et al.[20].

Nevertheless, until recently the existing evidence indicated that the majority of particles, including fast alphas, undergoes mixing during a sawtooth crash, so that their density profiles are flattened in the relaxation domain (unless, of course, the profile had already been broad before the crash[21]). In a 2014 review[22], it was concluded that "the effects of kink modes on fast ions seem to be understood." This was followed by the construction of computationally efficient reduced models, which are used in integrated transport simulations to make quantitative predictions for experiments such as ITER, assuming that all sawtooth crashes have the same effect[23].

It turns out, however, that the $q$ profile can have a significant influence on the transport of alpha particles. Using a heuristic model of a sawtooth crash, Jaulmes et al.[24] simulated the redistribution of alpha particles in a JET-like configuration, showing that the energy threshold for decoupling can be expected to lie at a few

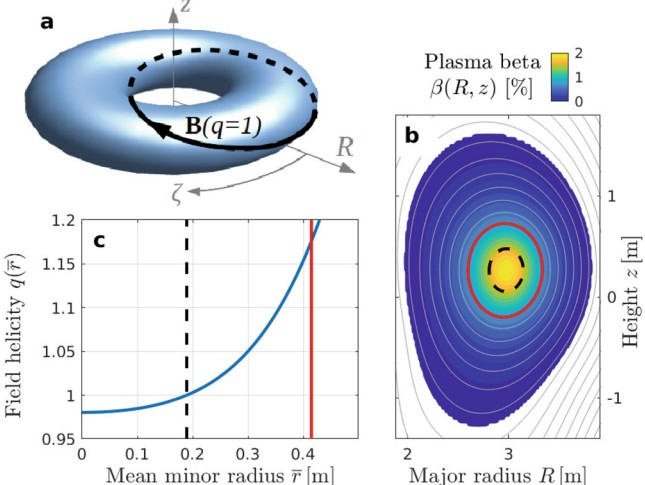

**Fig. 1 Plasma model based on the JET tokamak. a** Toroidal geometry in cylinder coordinates $(R, z, \zeta)$. The black curve represents a magnetic field line with helicity $q = 1$. **b** Shape of the plasma cross-section in a poloidal $(R, z)$ plane. The gray contours are magnetic flux surfaces. The color contours show the plasma beta, which measures the ratio of thermal to magnetic pressure as $\beta = 2\mu_0 P/B_0^2$ with $\mu_0 = 4\pi \times 10^{-7}$ H m$^{-1}$. **c** Central portion of a field helicity profile $q(\bar{r})$ that facilitates energy-selective alpha particle confinement. The dashed black line marks the $q = 1$ surface. The red line is the boundary of a reduced simulation domain.

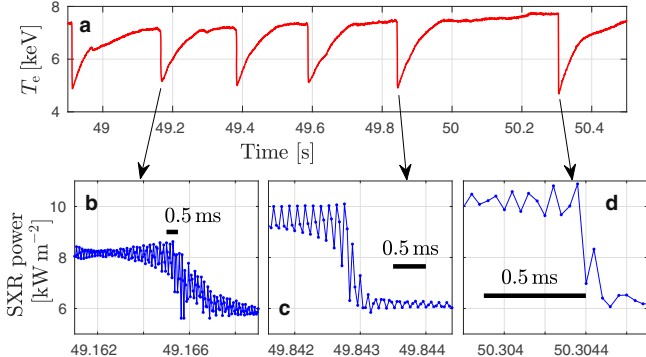

**Fig. 2 Examples of sawtooth crashes in JET.** Measured in the core of plasma pulse 95679[5,6]. **a** Evolution of the electron temperature $T_e$ inferred from electron cyclotron emission (ECE) sampled at 1 ms intervals. **b–d** Soft x-ray (SXR) signal showing details with a 40 µs sampling rate.

100 keV for trapped, and above 1 MeV for passing alpha orbits when $|1 - q|$ is sufficiently small. The important point to note then is this: When the magnitude of the parameter $|1 - q|$ drops to the level of a few percent, the resonance condition between alpha particles and the kink becomes sensitive to drifts associated with magnetic gradients ($\nabla B$ and curvature) even for passing alpha particles. This effect was not considered in the existing theory and it is the key insight underlying the present study.

In this work, using kinetic-MHD hybrid simulations, we show that there exists a parameter window where MeV-class alphas can sustain a sharply peaked density profile even inside the sawtooth mixing radius, whereas the majority of partially slowed-down alphas with energies of a few 100 keV or less are strongly mixed. This confirms earlier predictions of the heuristic models. The observations are explained in terms of a synergistic effect that emerges for a type of sawteeth where the magnetic field helicity remains close to unity ($q \sim 1$). An optimal crash time scale facilitates detuning of fast alphas from the internal kink, while helium ash remains phase-locked. However, detuning due to rapid parallel streaming along the **B** field is effective only until the particles enter a resonant reconnection layer, which is why electron profiles are eventually flattened. It is then the magnetic-drift-induced difference between orbit topology and magnetic topology that allows the majority of fast alphas to sustain a peaked density profile throughout the reconnection process. This realizes the looked-for energy-selective confinement for all pitch angles.

## Results

**Numerical simulation of a reconnecting internal kink.** Our simulations are performed using a so-called hybrid code[25,26], which solves visco-resistive magnetohydrodynamic (MHD) equations for the bulk plasma and kinetic equations for the fast ion minority species, whose inertia is assumed to be negligible compared to that of the bulk plasma. The plasma size and field strength are based on JET. The magnetic axis has a major radius of $R_0 = 3$ m and a field strength of $B_0 = 3.7$ T. The plasma current is $I_p = 2.5$ MA. The exact plasma composition is irrelevant in our single-fluid MHD model; here, the chosen bulk ion density and effective particle mass yield a central Alfvén speed of $v_{A0} \approx 8 \times 10^6$ m$s^{-1}$ (for pure deuterium, this corresponds to a central density of about $5.3 \times 10^{19}$ m$^{-3}$). Figure 1a shows the plasma torus schematically in right-handed cylinder coordinates $(R, z, \zeta)$. Figure 1b shows the plasma cross-section in the poloidal $(R, z)$ plane, where the toroidal angle $\zeta$, magnetic field vector **B** and plasma current are all pointing out of the plane. Figure 1c shows the chosen profile of the field helicity $q$, which we treat as a free parameter.

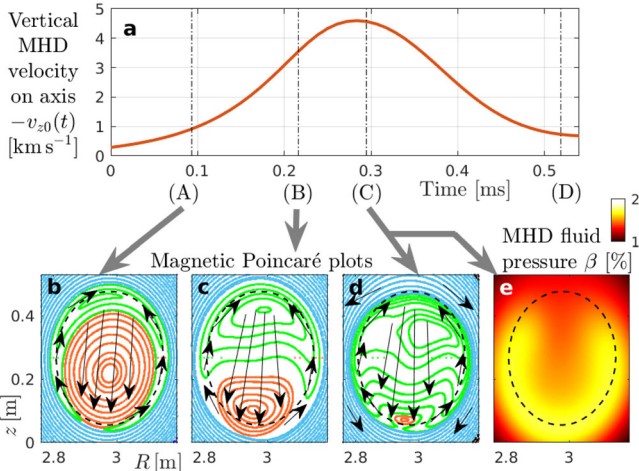

**Fig. 3 Simulated sawtooth crash.** Starting from the equilibrium in Fig. 1. **a** Evolution of the vertical component of the MHD velocity $v_{z0} = v_z(R_0, z_0, \zeta_0)$ evaluated at the plasma center (magnetic axis) at $R_0 = 3$ m, $z_0 = 0.27$ m, $\zeta_0 = 0$. **b–d** Poincaré plots taken in the poloidal $(R, z)$ plane at $\zeta_0$, showing the topology of the magnetic field for the snapshots labeled (A), (B) and (C), whose times 0.1 ms, 0.2 ms and 0.3 ms are measured from the instant where macroscopic displacement becomes visible. Red Poincaré contours have helicities $q < 1$. Green islands and blue periphery have $q > 1$. Arrows roughly indicate **E** × **B** flow directions. See Supplementary Figs. 3 and 5 for further details. **e** Contour plot of the bulk plasma beta $\beta(R, z, \zeta_0)$ at 0.3 ms, snapshot (C). Here, $\beta$ behaves as an MHD fluid. Its initial form was shown in Fig. 1b. The dashed circle is the initial $q = 1$ surface.

For the purpose of this study, it is not necessary to simulate the reconnection process in all its multi-scale detail, which is extremely challenging even with modern supercomputers[27]. Considering the type of sawtooth that is associated with an internal kink instability and proceeds in a fashion similar to that envisioned by Kadomtsev[4] and Wesson[28], it suffices here to simulate the associated global changes in the magnetic topology and the global electric drift. We require that the plasma is mixed in a similar volume and on a similar time scale of a few 100 µs as in experiments (Fig. 2). By matching the crash time scale and the size of the relaxing domain, we can match the speed of the displacement.

The size of the relaxing domain was inferred from electron temperature measurements like those in Fig. 2, which determine the approximate location of the $q = 1$ surface that is indicated by dashed lines in Fig. 1. The relaxation region is relatively small, so that it suffices in many of our simulations to cover only the inner 25% of the plasma's magnetic flux space. In that case, an artificial non-slip boundary is placed along the red line in Fig. 1, at about twice the sawtooth mixing radius. Inside that region we choose the $q$ profile to be close to unity and relatively flat, which yields a configuration that is both numerically tractable and practically relevant for long-pulse scenarios in future ITER experiments.

Figure 3a shows the evolution of the vertical ($z$) component of the local MHD velocity $v_{z0}(t)$ measured at the magnetic axis, which consists primarily of electric drift, $\mathbf{v}_E = \mathbf{E} \times \mathbf{B}/B^2$. Its magnitude reaches nearly 5 km$s^{-1}$, or 0.06% of the Alfvén speed. Here the sign of $v_{z0}$ is negative, but we have reversed it in Fig. 3a to visualize the growth, saturation and decay of the internal kink mode.

Magnetic reconnection converts magnetic to kinetic energy. The kinetic energy carried by the flow in Fig. 3a increases exponentially at first and saturates when the reconnection stops. Here, the reconnection phase lasts about 0.3 ms as one can see in Fig. 3b–d, where we show Poincaré plots of the magnetic field

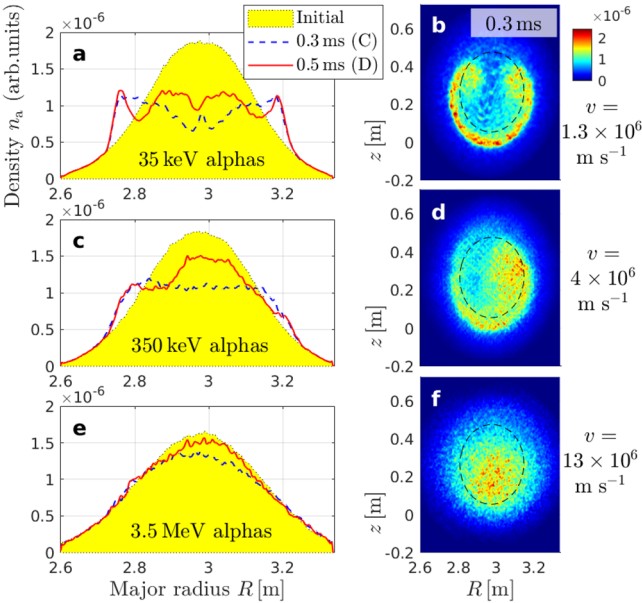

**Fig. 4 Spatial transport of alpha particles.** During the sawtooth crash in Fig. 3, we followed particles initialized with different kinetic energies $K$: **a**, **b** 35 keV, **c**, **d** 350 keV, **e**, **f** 3.5 MeV. The left column shows, in arbitrary units, snapshots of the radial density profile $n_a(R)$ at the height of the midplane ($z_0 = 0.27$ m), integrated over $\zeta$ and all pitch angles. On the right, we show the density field $n_a(R, z, \zeta_0)$ in the poloidal plane around the toroidal angle $\zeta_0 = 0$ at the time of snapshot (C) 0.3 ms. The particle velocities $v = \sqrt{2K/M_a}$ are shown for reference. These data were obtained in simulations of the reduced domain encircled by a red line in Fig. 1b. Very similar results are obtained for the full domain as shown in Supplementary Fig. 3c.

topology during that period. The structures in Fig. 3 should be imagined as winding helically around the plasma center, like the field line sketched in Fig. 1a. In the $\zeta_0 = 0$ plane that we use for all our Poincaré and contour plots in Fig. 3, the electric drift (black arrows) advects the central region of the plasma downward and returns upwards along the sides. Magnetic flux surfaces are torn and reconnected at the bottom of the plot, where the original plasma core with $q < 1$ (red contours) shrinks and nearly vanishes around the time of snapshot (C) at 0.3 ms. The field helicity in the region with green contours is slightly above unity ($q \gtrsim 1$) and nearly uniform, so the green structures appearing in the Poincaré plots may be ignored.

The dynamics seen here resemble a reconnecting internal kink mode as envisioned in the Kadomtsev model[4], which has been observed in well-diagnosed experiments[29], although other behavior is possible (e.g., refs. [30–34]). The pair of convective cells associated with the internal kink redistributes the bulk plasma (modeled as an MHD fluid) in such a way that the pressure peak of Fig. 1b acquires a horseshoe-like structure at the end of the crash as shown in Fig. 3e[35] (we suspect that the associated changes in the MHD force balance are responsible for the compression of the reconnected core in Fig. 3d).

While the $\mathbf{E} \times \mathbf{B}$ flows decay after snapshot (C), they cause further interchange and mixing as described by Wesson[28] for about 0.1 ms. As the flows weaken towards the end of the simulation ($\gtrsim 0.5$ ms), their effect is eventually overcome by sources, which tend to restore the original plasma current profile in our simulation model.

**Alpha particle transport**. The sawtooth crash in our simulation lasts less than 0.5 ms. This lies in the range of experimentally observed crash times $\tau_{crash} \sim 0.1 \ldots 2$ ms in Fig. 2, so we take this to

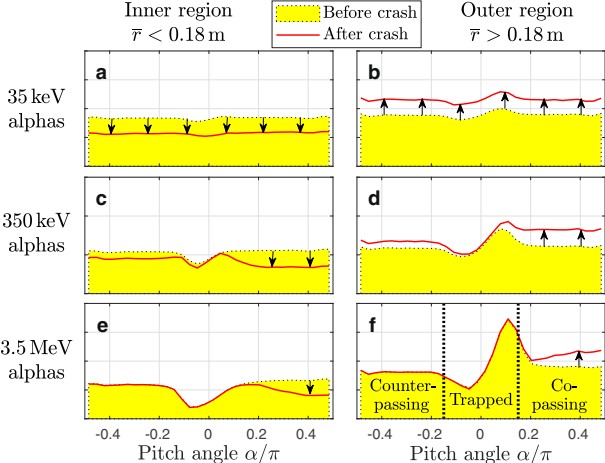

**Fig. 5 Pitch angle distributions before and after the crash.** Alpha particle tracers were loaded in the whole range of pitch angles defined by $\alpha = \sin^{-1}(v_\parallel/v)$ with initial energies **a**, **b** 35 keV, **c**, **d** 350 keV, **e**, **f** 3.5 MeV. These data were obtained in simulations of the entire plasma in order to capture all orbits, including those near the trapped-passing boundary, which traverse both the plasma core and the periphery[36]. Spatial integration was performed separately for the inner region $\bar{r} < 0.18$ m (left) and the outer region $\bar{r} > 0.18$ m (right), whose border is near the initial $q = 1$ radius $\bar{r}_1 \approx 0.19$ m. More complete views of the velocity distributions are shown in Supplementary Figs. 7 and 8.

be a meaningful scenario to study the redistribution of fast alphas and helium ash during such a relaxation event in JET geometry.

We fill our simulation domain with mono-energetic alphas populating all pitch angles, except for the loss cone[36]. The initial radial profiles of the particle density at the height of the midplane ($z_0 = 0.27$ m) are shown as yellow-shaded areas in Fig. 4. The widths of the profiles are chosen to be similar to the neutron emission profile observed in JET[5,6,37], but vary somewhat due to the energy-dependence of magnetic drifts and gyroradii. Our simulations are run with a negligibly low alpha particle density to ensure that our alphas remain passive, so that the bulk plasma dynamics are identical in all cases.

Figure 4a, c, e shows the evolution of the density profile of alpha particles with kinetic energies $K = M_a v^2/2 = 35$ keV, 350 keV and 3.5 MeV during the simulated sawtooth crash. Around the time of snapshot (C) at 0.3 ms, where the MHD flows are strongest, the central density of 35 keV alphas is temporarily reduced to 40% of its initial value. It recovers partially during the aftermath of the sawtooth crash and settles at about 60% of its initial value around the time of snapshot (D) at 0.5 ms, which can be regarded as the relaxed state. In contrast, the 3.5 MeV alpha density remains centrally peaked, and recovers almost fully after a small temporary reduction. Intermediate behavior is seen at intermediate energies (see also Supplementary Fig. 8).

The redistribution of particles at different pitch angles $\alpha = \sin^{-1}(v_\parallel/v)$ with $v_\parallel/v \equiv \mathbf{v} \cdot \mathbf{B}/(vB)$ is shown in Fig. 5. Co-passing particles occupy roughly the domain $\alpha \gtrsim 0.15\pi$, counter-passing particles have $\alpha \lesssim -0.15\pi$, and particles trapped by the magnetic mirror force are found around $-0.15\pi \lesssim \alpha \lesssim 0.15\pi$. Here, the attributes 'co' and 'counter' refer to the direction of the plasma current, which in the present case coincides with the direction of $\mathbf{B}$. The initial particle distributions (yellow-shaded areas in Fig. 5) have been prepared to be as uniform in pitch as is physically possible. The nonuniformity around $\alpha \approx 0$ that increases with energy $K$ is an inevitable consequence of the sharply peaked density profile, magnetic drifts and the presence of many non-standard orbits[36].

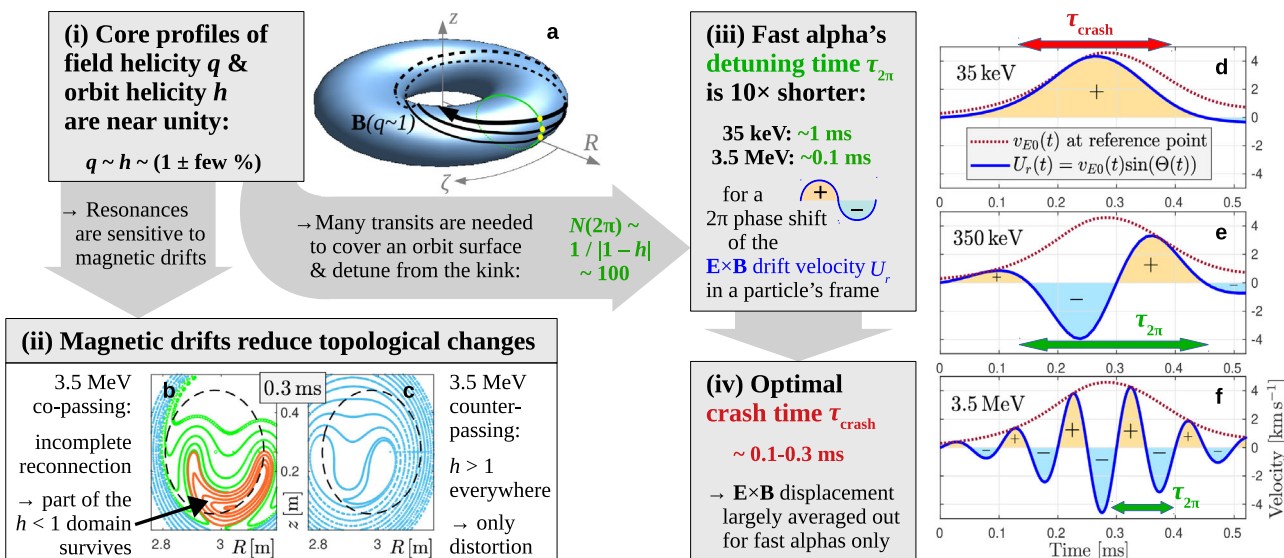

**Fig. 6 Physics of energy-selective confinement of alpha particles during a sawtooth crash. a** Toroidal magnetic surface with field helicity $q \sim 1$, where **B** field lines nearly close after each toroidal turn, so that they cover the surface only after a large number $N(2\pi)$ of turns. **b, c** Poincaré plots showing the perturbed orbit topology of co- and counter-passing 3.5 MeV alphas during the sawtooth crash in Fig. 4. **d-f** Schematic evolution of the electric drift velocity associated with the internal kink mode in Fig. 4, measured in the frame of reference that moves with passing alpha particles of different kinetic energies.

Results for the slow 35 keV alphas are plotted in Fig. 5a, b, where one can see that the sawtooth crash causes net outward displacement of nearly uniform magnitude across the entire range of pitch angles $-\pi/2 \leq \alpha \leq \pi/2$. At 350 keV in Fig. 5c, d, about 40% of the co-passing alphas are displaced, whereas counter-passing particles are only weakly perturbed, and mirror-trapped particles even less. In the case of 3.5 MeV in Fig. 5e, f, net transport is observed only in the domain of co-passing particles, whose number in the inner domain decreases by about 20%. The small reduction in the overall density in Fig. 4e is due to these displaced co-passing particles.

**Physical mechanism.** A schematic illustration of the factors responsible for the energy-selective confinement of fast alphas in our simulations is shown in Fig. 6. Before going into the details, here is a quick run-through: The first key factor, represented by box (i) in Fig. 6, is that the field helicity $q$ is close to unity. This has the consequence that (ii) alpha particle resonances with respect to the internal kink are sensitive to magnetic drifts, and (iii) only fast alphas have a resonance detuning time $\tau_{2\pi} \sim 0.1$ ms that is shorter than (iv) the sawtooth crash time $\tau_{\text{crash}} \sim 0.3$ ms in Fig. 3. Factors (i), (iii) and (iv) were anticipated by theoretical analyses in ref. [19], as was the fact that the threshold energy is lower for trapped particles than for passing ones (Fig. 5). The reason for the observed difference between co- and counter-passing particles in Fig. 5, however, remained elusive[24,38]. Here, the physical picture is completed by including (ii) the magnetic-drift-induced shift of the resonances. The combination of factors (i)–(iv) in Fig. 6 facilitates selective confinement of fast alphas and mixing of slow alphas. Let us elucidate this synergism in detail.

The condition for a charged particle to resonate with the internal kink can be expressed as $h \equiv \omega_{\text{tor}}/\omega_{\text{pol}} = 1$, where the orbit helicity $h \approx \bar{r}v_{\text{tor}}/(R_0 v_{\text{pol}})$ is the ratio of toroidal and poloidal transit frequencies and can be thought of as the kinetic counterpart of the field helicity $q \approx \bar{r}B_{\text{tor}}/(R_0 B_{\text{pol}})$ modified by the combined effect of magnetic drifts $\mathbf{v}_{\text{d}}$ and the mirror force[39–42]. The radial excursion caused by the magnetic drift is given by $(\Delta R)_{\text{d}} \approx v_{\text{d}}/\omega_{\text{pol}} \approx v_{\parallel}[1 + v_{\perp}^2/(2v_{\parallel}^2)]M/(ZeB_{\text{pol}}\bar{r})$, so it is proportional to a particle's

mass-to-charge ratio $M/(Ze)$ and velocity $v$, and inversely proportional to the plasma current[43]. Co-(counter-)passing orbits are shifted out-(in-)ward in major radius $R$.

Figure 7 shows several examples of orbit helicity profiles $h$ as functions of the radial distance $X = R - R_0$ from the plasma center. $h(X)$ profiles are shown for co- and counter-passing alpha particles with energies $K = 35$ keV, 350 keV and 3.5 MeV, and velocity pitch $v_{\parallel}/v = \sin(\pm 0.48\pi) \approx \pm 1$. One can see that, with increasing kinetic energy $K$, the orbit helicity profiles $h$ deviate increasingly from the field helicity profile $q \approx h(35 \text{ keV})$. The mirror force causes particles to spend more time in regions of smaller $R$, where the field is stronger, so that the orbit helicity $h$ is reduced for the outward-shifted co-passing ions, and increased for inward-shifted counter-passing ions. Figure 7 shows that in the present case these effects are large enough to entirely eliminate the $h = 1$ resonance for counter-passing 3.5 MeV alphas by raising their $h$ profile above unity everywhere.

This difference between the field helicity $q$ and the orbit helicity $h$ has the consequence that magnetic drifts reduce the amount of reconnection that occurs in the topology of fast alpha particle orbits as highlighted in box (ii) of Fig. 6: Fig. 6b shows that reconnection in co-passing orbit topology is incomplete, so that a large region with $h < 1$ survives through the crash. For counter-passing orbits in Fig. 6c, we have always $h > 1$ everywhere, which means that an $h = 1$ resonance never forms and the orbit topology is merely distorted by the magnetic perturbations associated with the internal kink. More details about the orbit topology evolution can be found in Supplementary Fig. 5.

This mechanism explains the unresponsiveness of counter-passing alphas and the partial mixing of co-passing alphas in Fig. 5. However, it must be noted that the observed amount of co-/counter-passing asymmetry is linked to the choice of the boundary between the inner and outer regions, which we have placed near the initial $q = 1$ radius $\bar{r}_1 \approx 0.19$ m. For instance, counter-passing particles in the intermediate case with 350 keV are also subject to mixing, but this is only partially visible in Fig. 5c, d because their mixing radius is smaller than the $q = 1$ radius, as is evident from Fig. 7. Results for a smaller inner region ($\bar{r} < 0.06$ m) can be found in Supplementary Fig. 8c–e.

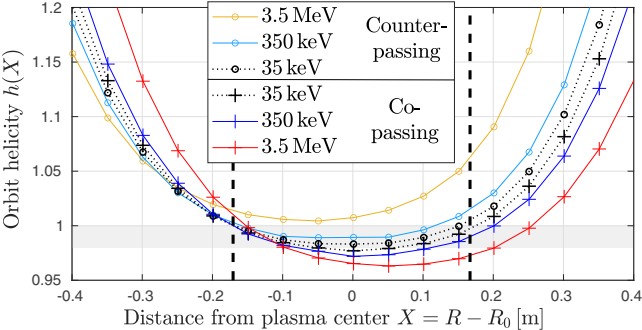

**Fig. 7 Initial orbit helicity profiles.** This figure shows how $h = 1$ resonances shift or even vanish as the orbit helicity $h$ of alpha particles increasingly deviates from the field helicity $q$ with increasing kinetic energy. Profiles of the orbit helicity $h(X) = \omega_{tor}/\omega_{pol}$ are plotted as a function of the radial position $X = R - R_0$ where an alpha particle orbit crosses the midplane $z_0 = 0.27$. The $q$ profile is not shown, but it lies between the two black curves for $h(35 \, \text{keV})$. The dashed vertical lines are the $q = 1$ radii. Orbits with initial helicities in the range $\min\{q\} \lesssim h \lesssim 1$ (shaded area) and nearby are likely to be subject to resonant interaction and reconnection some time during the sawtooth crash (an accurate prediction may be made using Hamiltonian analysis in helical coordinates as in Fig. 8.12 of ref. [2]). The pitch dependence of the outer $h = 1$ radius at 3.5 MeV is shown in Supplementary Fig. 6.

Let us now proceed to the next physical factor. The second implication of having $q \sim h \sim 1$ in box (i) of Fig. 6 is that field and orbit trajectories nearly close on themselves after one toroidal transit as illustrated schematically in Fig. 6a. Taking a concrete value for illustration, say $h \sim 1 \pm 0.01$, this means that a particle has to perform on the order of $N(2\pi) \sim 1/|1-h| \sim 100$ toroidal transits before its trajectory covers a toroidal orbit surface. When viewed from above, the motion illustrated schematically in Fig. 6a resembles the slow apsidal precession of a planetary orbit around the Sun.

During these $N(2\pi)$ transits, the phase of the radial $\mathbf{E} \times \mathbf{B}$ drift velocity $U_r(t)$ measured in the frame of reference moving with a chosen particle will also complete a $2\pi$ phase shift. This is illustrated in Fig. 6d–f. In our simplified example, we assume that the electric drift in the moving frame has a sinusoidal form $U_r(t) = v_{E0}(t)\sin(\Theta(t))$. For the envelope $v_{E0}(t)$ we chose the vertical MHD velocity from Fig. 3a as $v_{E0} = -v_{z0}$. The evolution of the phase is modeled as $\Theta(t) = 2\pi t/\tau_{2\pi}$, where $\tau_{2\pi} = \tau_{tor}N(2\pi) \sim \tau_{tor}/|1-h|$ is the time scale for the kink mode's phase to slip by $2\pi$. Thus, $\tau_{2\pi}$ plays the role of a resonance detuning time.

As indicated in box (iii) of Fig. 6, the resonance detuning times of newly born 3.5 MeV alphas and 35 keV helium ash differ by a factor 10. Sawteeth with crash times $\tau_{crash}$ of a few 100 µs as in Fig. 3 are then shorter than the 1 ms detuning time of 35 keV alphas and longer than the 0.1 ms detuning time of 3.5 MeV alphas in our setup. This leads to the final box (iv) of Fig. 6: only for sufficiently fast alphas the net $\mathbf{E} \times \mathbf{B}$ displacement is likely to vanish through the cancellation of positive and negative peaks of $U_r(t)$ as in Fig. 6f. Depending on the initial phase, there is still a 50% chance for cancellation at 350 keV in Fig. 6e, where $\tau_{2\pi} \sim \tau_{crash}$. In contrast, the 35 keV alphas in Fig. 6d typically remain in phase with the kink's electric field for the entire duration of the crash. This explains why the density field of slow 35 keV alphas in Fig. 4b develops the same horseshoe-like structure as the MHD fluid in Fig. 3e. At the intermediate energy of 350 keV in Fig. 4d, the density field has a swirling tear-drop structure as the poloidal spreading competes with the displacement due to the $\mathbf{E} \times \mathbf{B}$ drift on a similar time scale. At 3.5 MeV, the toroidal speed is so high that poloidal spreading outpaces

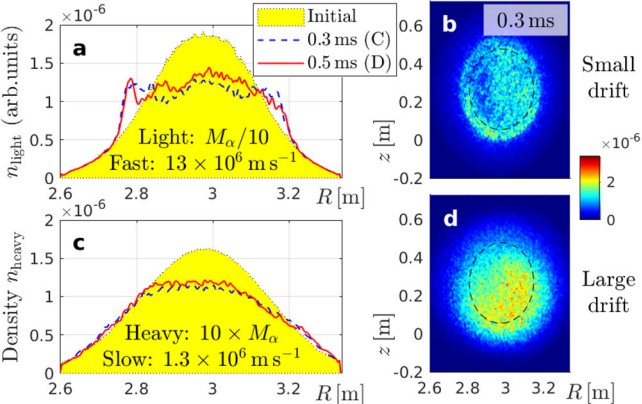

**Fig. 8 Spatial transport of light and heavy particles.** This numerical experiment isolates the effects of high velocity (**a**, **b**) and large magnetic drift (**c**, **d**). Arranged like Fig. 4.

$\mathbf{E} \times \mathbf{B}$ drifts. This detuning (phase slippage) combined with the magnetic-drift-induced resonance shift described earlier allows the density field of 3.5 MeV alphas in Fig. 4f to maintain a compact peak with only a minor and temporary helical distortion.

It can be verified that all ingredients are needed. For instance, the rapid motion of thermal electrons allows them to satisfy the detuning condition (iii) and the scale separation condition (iv) in Fig. 6. However, the lack of magnetic drifts causes electrons to stick closely to magnetic field lines and undergo strong mixing during sawtooth crashes that involve magnetic reconnection. This is confirmed in Fig. 8a, which shows the redistribution of an artificial particle species with charge number $Z = 2$, reduced mass $M_{0.1} = 0.1 \times M_a$ and high speed $v = 13 \times 10^6 \, \text{m s}^{-1}$. Its magnetic drift is as small as that of slow 35 keV alphas with mass $M_a$, while the transit frequency is as high as that of fast 3.5 MeV alphas. One can see that the density profile in Fig. 8a is flattened.

Figure 8c shows that the density profile is also flattened in the opposite limit, where heavy particles ($M_{10} = 10 \times M_a$, $Z = 2$) travel at low speed ($v = 1.3 \times 10^6 \, \text{m s}^{-1}$). Here, the magnetic drift (and gyroradius) is as large as that of fast 3.5 MeV alphas with mass $M_a$, so that the density field in Fig. 8d is similarly blurred as in Fig. 4f. Nevertheless, the density profile in Fig. 8c is subject to much stronger flattening than in Fig. 4e. This must be due to the particle speed $v$ being smaller by a factor 10 since all other parameters are identical.

The results in Fig. 8 thus show that the resonance shift due to the magnetic drift, which (for a given $v$) is larger for particles with smaller charge-to-mass ratio $Ze/M$, is not sufficient but necessary for preventing profile flattening. Similarly, detuning due to rapid parallel streaming is necessary but not sufficient. This proves that the selective confinement of fast alphas in a wide range of pitch angles is realized only through the combination of the four factors summarized in Fig. 6.

## Discussion

Simulations of a large tokamak plasma based on JET confirmed the existence of a parameter window where the majority of MeV-class alpha particles can remain well-confined in the plasma core during a benign sawtooth crash that strongly redistributes less energetic ions and electrons. We proposed a physical picture that extends the existing theory[19] by accounting for the modification of alpha particle resonances[44–47] and orbit topology via magnetic drifts. This explains the observed differences between the responses of co- and counter-passing alphas during a sawtooth crash[24,38,47]. A reduced model that captures these effects is now available[48].

The illustration in Fig. 6 is of course simplified in order to highlight the primary mechanisms and their synergistic effect in a clear manner. For instance, the transit number $N(2\pi)$ that determines the detuning time $\tau_{2\pi}$ actually varies with particle energy and radial location as one can readily infer from the orbit helicity profiles $h(X)$ in Fig. 7. These orbit helicity profiles also evolve in time, with a tendency to rise during the crash. Thus, the resonances are dynamic: their location, width and their very existence evolve rapidly. Moreover, the $\mathbf{E} \times \mathbf{B}$ flow pattern is nonuniform in space. Along particle orbits that perform large magnetic drifts, the direction of the electric drift may thus vary even during a single transit. This influences the effective magnitude of the electric drifts for fast alphas. Our simulations capture these complexities accurately, while also confirming the basic physical picture drawn in Fig. 6.

The insights won are of interest for the tokamak-based fusion reactor R&D programs that are currently pursued around the world. Those R&D activities would benefit from the possibility of using sawteeth for removing helium ash and other impurities without deteriorating fast alpha confinement. All else being similar, the time scales summarized in Fig. 6 would be 2–3 times longer in ITER and DEMO reactors due to their larger major radii $R_0 \approx 6$ m and 9 m, respectively. The scenario we studied hence lies quantitatively in the right 'ball park'. Thus motivated, we conclude this study with a preliminary discussion about the practicality of the method.

Overall, it has become clear that the parameter window of interest is narrow. This poses several practical obstacles. First, since the magnetic drifts play an important role, the mechanism described here may be utilizable only when the plasma current is not too high. The advantages and disadvantages of this operational regime will have to be weighed. Potential applications in spherical tokamaks may be worthy of consideration[49–51].

Second, the theoretical evidence implies that the redistribution of the alpha particles is sensitive to the pre-crash profile of the field helicity $q$. This prediction is corroborated by preliminary results of $q$ profile scans reported in Supplementary Figs. 9–14, which indicate that the fast alpha profiles undergo significant flattening when $|1 - q| \gtrsim \mathcal{O}(5\%)$ for the considered plasma parameters ($I_{\mathrm{p}} = 2.5$ MA, $B_0 = 3.5$ T). In order to experimentally validate and utilize the sawtooth-mediated energy-selective mixing and confinement in a reactor, it is thus necessary to ensure that only certain types of sawteeth occur, namely those for which the field helicity remains close to unity ($q \sim 1$). This requires more accurate $q$ profile diagnostics, a better understanding of sawtooth physics, and more precise plasma control schemes, where one controls not only the sawtooth period but also the form of the crash. The benefits of advancing these capabilities go far beyond the subject of the present work: Besides enabling alpha particle density control, another important advantage of operating reliably in a regime with benign sawteeth and $q \sim 1$ is that it minimizes the forced formation of magnetic islands in the outer core and peripheral plasma (see Supplementary Fig. 4), which is important for maximizing performance and avoiding plasma disruptions.

Other factors requiring consideration are the bulk plasma confinement in regions where $q \sim 1$, plasma rotation, and the role of pre- and post-cursor oscillations. While important for sawtooth physics, it must be noted that the helical distortions observed as pre- and postcursor oscillations in a rotating plasma do not necessarily affect overall confinement. On the collisionless time scales where adiabatic invariants of guiding center motion are valid, confinement can be broken only by resonant interactions[52,53] via convective phase space instabilities[54–56], resonance overlaps[57], or reconnection phenomena (as in the present work).

It is necessary to quantify the stabilizing or destabilizing effect that fast alphas in realistic concentrations exert on the internal kink and other MHD modes[7,18,58–60]. The same counts for the influence of sawtooth-induced alpha particle transport on the background plasma from the viewpoints of heating, current drive and plasma rotation. Investigations in this direction are motivated by the local imbalance between co- and counter-passing alphas that was caused by the sawtooth crashes in our simulations. This effect can be significant even for larger crashes triggered at $|1 - q| \sim \mathcal{O}(10\%)$ and may be a subject of interest on its own. It can influence the evolution of the $q$ profile by adding negative and positive plasma current, and it may exert a sheared toroidal torque around the radius of the pre-crash $q = 1$ surface. This, in turn, can lead to complex nonlinear feedback whose consequences deserve further study.

Last but not least, another potential obstacle becomes evident if one considers not only a single sawtooth crash but multiple sawtooth cycles. Since a sawtooth crash is a mixing process and, thus, does not affect an already flat profile, it is clear that the mechanism we have described becomes noticeable only if the density profiles of the fusion-born alphas and resulting ash are peaked within the mixing radius. The formation of such a peaked profile in the density of fusion-born alphas requires that the D-T fusion fuel has a peaked pressure profile in the first place. The sustained core-localized alpha heating can help to recover this condition after a sawtooth crash, but the interval between successive sawteeth must be sufficiently long to allow for the recovery of a sharply peaked bulk plasma pressure. Although it is still unclear how to realize this situation in practice, the observation of impurity accumulation in tokamak plasmas with enhanced overall confinement suggests that there exists a density pinch mechanisms that can lead to central peaking and requires further study. Techniques to deliver fusion fuel deep into the plasma core would also help and constitute yet another challenging topic of fusion research.

On a positive note, the practical considerations and the physics we have discussed exemplify the rich nonlinear dynamics that magnetically confined fusion plasmas support. We expect that successful ITER experiments have not only the potential to yield valuable insights needed for fusion power plants; ITER is also likely to inspire new experiments for burning plasma physics studies, with sawtooth crashes and the associated reconnection in magnetic and orbit topology playing a prominent role. The parameter window of interest that we identified in this work — namely, a field helicity profile $q(\bar{r})$ close to unity — lies precisely in the regime envisioned for ITER long-pulse operation. At the same time, this regime with $q \sim 1$ still exhibits unsolved mysteries and, thus, opportunities (e.g., refs. [33,61–64]). The effects discussed here and further research should throw more light on these matters.

## Methods

**Simulation model and parameters**. A detailed description of the model equations and numerical techniques along with benchmarks, convergence tests, sensitivity studies and experimental validation for different types of fast-ion-driven instabilities in tokamaks can be found in the literature[65–70]. The simulation setup used in the present work was described in ref. [36] and further details including sensitivity tests are provided in the Supplementary Information accompanying this paper. For the reader's convenience, the following paragraphs provide a summary of the numerical parameters used in the simulations and data analysis. The relevant plasma parameters are summarized in Table 1.

**Discretization, dissipation, filtering**. Our cylindrical mesh in the reduced simulation domain bounded by the red circle in Fig. 1b consists of $N_R \times N_z \times N_\zeta = 220 \times 220 \times 96$ points; very similar results were obtained with $520 \times 520 \times 96$ and $220 \times 220 \times 192$. Simulations for the full domain used $520 \times 520 \times 96$ grid points. The 4th-order Runge–Kutta time step was about 1 ns for the MHD solver, which is constrained by the Courant–Friedrichs–Lewy (CFL)

**Table 1 Plasma simulation parameters.**

| | |
|---|---|
| Major radius | $R_0 = 3.0$ m |
| Mean minor radius | $\langle a \rangle \approx 1.2$ m |
| Plasma current | $I_p = 2.5$ MA |
| Toroidal field strength | $B_0 = 3.7$ T |
| Thermal/magnetic pressure ratio | $\beta_0 = 2\%$ |
| Safety factor (magnetic axis) | $q_0 = 0.98$ |
| Safety factor (plasma edge) | $q_{edge} = 5.44$ |
| Number density (bulk, deuterium) | $n_{b0} = 5.3 \times 10^{19}$ m$^{-3}$ |
| Number density (alpha particles) | $n_{a0} \lesssim 10^{-7} \times n_{b0}$ |

Partly based on JET pulse number 95679[5,6]. The pressure excludes fast ions. The $q$ profile in the central core was chosen such that $q_0$ - 1. The alpha particle density in the simulations was initialized with a negligibly low value, so that they behaved as passive tracer particles that did not affect the evolution of MHD modes. The subscript '0' indicates that a quantity was measured at the center (initial magnetic axis) of the plasma.

condition for fast magnetosonic waves. The value of the specific heat ratio that controls the plasma compressibility in our MHD simulations was fixed at $\Gamma = 5/3$. Larger time steps can be used for the alpha particles, because our setup (Table 1) does not rely on temporal averaging of particle-in-cell (PIC) noise: at 35 keV it suffices to advance particles with a 40 times greater time step, whereas 3.5 MeV alphas were pushed every 4th MHD time step. Test runs with more frequent pushing produced the same results.

The number of simulation particles representing the guiding centers of mono-energetic populations of alpha particles lay in the range $(1.5\ldots4) \times 10^6$ in the reduced domain and $(8\ldots22) \times 10^6$ in the full domain. A full-domain simulation with alphas distributed uniformly over the energy range $(0.35\ldots3.5)$ MeV as shown in Supplementary Fig. 8 used $37 \times 10^6$ simulation particles.

The values of the MHD coefficients controlling resistive, viscous and thermal diffusion are $\eta/\mu_0 = \nu = \chi = 10^{-6}\nu_{A0}R_0$, which is considered to be a reasonable compromise between numerical and physical considerations[68]. In the full domain, we filtered out toroidal harmonics $\exp(in\zeta)$ with $|n| > 12$ to suppress numerical instabilities. Such filtering was also necessary in $q$ profile scans where we simulated 50% of the magnetic flux space (see Supplementary Figs. 9–14), but it was not necessary when simulating only the inner 25% of the magnetic flux space for our default case with $q_0 = 0.98$.

**Gyroaveraging**. Our simulations were run with 4-point gyroaveraging around a particle's guiding center[68]. Very similar results were obtained in the zero-Larmor-radius (drift-kinetic) limit, even for 3.5 MeV alphas.

The reason for why the finite Larmor radius (FLR) has no notable effect here is that the fast alphas in our simulations were already well-confined without gyroaveraging, while the slow alphas that underwent strong mixing have small gyroradii.

It is also worth noting that gyroaveraging and magnetic drifts work in different ways. Gyroaveraging occurs effectively instantaneously around a given guiding center position, so that this aspect of FLR has merely a local smoothing effect. It tends to reduce the effective magnitude of the electric field and, consequently, the electric drift velocity and energy transfer, but it hardly affects the resonances as such. In contrast, the magnetic drift displaces the entire guiding center orbit contour relative to the magnetic flux contours, so its effect is global. In combination with the mirror force, the magnetic drift can shift or even eliminate resonances (Fig. 7). The commonly held notion of a 'drift-orbit-averaging effect' can hence be misleading. In principle, the magnetic drift may even enhance the resonant coupling between an MHD mode and fast particles, because it can localize the interaction domain poloidally[71] and, thus, maximize phase locking (minimize phase shift), albeit at the expense of reducing the interaction time.

We expect that FLR effects such as gyroaveraging will be more important when the population of alpha particles is large enough to influence MHD modes, which was not the case here. The large gyroradii of fast alphas are expected to reduce the efficiency of resonant coupling with MHD modes[68].

**Poincaré analysis of field and orbit topology**. All Poincaré plots presented in this work — including Figs. 3 and 6 of the main article and Supplementary Figs. 4, 5, 10 and 11 — were obtained by tracing magnetic field lines or the guiding center orbits of alpha particles in the self-consistent perturbed magnetic field **B** of the hybrid simulation. Although they were present in the simulations, electric drifts were not included in the Poincaré analysis and the magnetic field was not evolved while the test particles traced out their orbits. This means that our Poincaré contours reflect only the instantaneous topology of the orbits, not the motion of particles in the original hybrid simulation of the sawtooth crash, where the magnetic configuration changes while a particle is tracing out a toroidal orbit surface. See also ref. [19] for a related discussion.

The test particles were launched from the outer midplane ($R > R_0$, $z = z_0 = 0.27$ m) and advanced using a 4th-order Runge–Kutta algorithm. For each initial position, we recorded at least 450 Poincaré sections in the poloidal $(R, z)$ plane at $\zeta_0 = 0$. The direction in which successive points appear in the Poincaré plot allowed us to identify whether the local value of the field helicity $q$ and orbit helicity $h$ was greater or smaller than unity. This information was then used to assign colors for each of the Poincaré contours: red/orange for $q, h < 1$ and blue/green for $q, h > 1$.

## Data availability

The JET experimental data is stored in the Processed Pulse File (PPF) system, which is a centralized data storage and retrieval system for data derived from raw measurements within the JET tokamak, and from other sources such as simulation programs. These data are fully available for the EUROfusion consortium members and can be accessed by non-members under request to EUROfusion. The experimental data for JET pulse 95679 that is shown in Fig. 2 can be identified as (a) KK3-TE24 and (b) SXR-H10-JETPPF (calibrated). The reference equilibrium reconstructed by TRANSP code modeling[37] is g95679_83-50.22554.eqdsk (time stamp: 2019/12/19, 19:59). The data of the model equilibria constructed using the code CHEASE[72], alpha particle distributions modeled using the code VisualStart[36], and the numerical data of MEGA simulations are stored on devices administered by National Institutes for Quantum Science and Technology (QST). These data may be obtained after establishing an official research collaboration agreement with QST. Numerical data that is shown in the figures and support the outcome of this study are available from the corresponding author upon reasonable request.

## Code availability

Further information concerning the hybrid code MEGA[25,26,73] and the version used in this work[65,68] can be made available by the corresponding author upon reasonable request. The source code may be obtained after establishing an official research collaboration agreement with National Institutes for Quantum Science and Technology (QST). Alternatively, one may contact the author of the original code, Prof. Yasushi Todo at National Institute for Fusion Science (NIFS) (todo@nifs.ac.jp).

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

## Acknowledgements

A.B. thanks Yasushi Todo (NIFS, Japan) for valuable support in connection with the code MEGA. Insightful discussions with Nikolai Gorelenkov, William Heidbrink and Yurii Yakovenko following the presentation of our results at the 17th IAEA TCM EPPI meeting (Dec. 2021) are thankfully acknowledged. The simulations reported here were carried out using the supercomputer JFRS-1 at Computational Simulation Centre of International Fusion Energy Research Centre (IFERC-CSC) in Rokkasho Fusion Institute of QST. Preliminary studies were also conducted using the supercomputer SGI ICE X in the Japan Atomic Energy Agency (JAEA). The work by S.S. was supported by Grants-in-Aid for Scientific Research from JSPS (Grant No. 20K14447). This work has been partially carried out within the framework of the EUROfusion Consortium and has received funding from the Euratom research and training programme 2014-2018 and 2019-2020 under Grant Agreement No. 633053. The views and opinions expressed herein do not necessarily reflect those of the European Commission.

## Author contributions

A.B., K.S., Y.O.K., V.G.K., P.L., S.S. and M.Y. contributed to the clarification of the physical mechanisms. Y.O.K., V.G.K., M.N., Ž.Š., J.G. and JET Contributors participated in the design, execution and analysis of JET experiments. J.G. and S.I. conceived the project, and Y.O.K. and A.B. identified the particular problem of interest. A.B. designed, performed and analyzed the simulations and wrote the manuscript.

## Competing interests

The authors declare no competing interests.

## Additional information

## JET Contributors

A. Bierwage [1,2 ✉], K. Shinohara [2,3], Ye.O. Kazakov [4], V. G. Kiptily[5,11], Ph. Lauber[6,11], M. Nocente [7,8], Ž. Štancar [5,9,11], S. Sumida[2] & J. Garcia [10,12]

A full list of members and their affiliations appears in the Supplementary Information.

