## [Peer Review File · Nature Communications]

Energy-selective confinement of fusion-born alpha particles during internal relaxations in a tokamak plasmaREVIEWER COMMENTS

Reviewer #1 (Remarks to the Author):

Summary and general comments:

A kinetic-MHD hybrid simulation is used to model alpha particle transport during a sawtooth relaxation event driven by an $m=1$, $n=1$ internal kink in a JET equilibrium. It is found that for certain parameters, namely q within a few percent of 1, the highest energy alphas ($E \sim 3.5$ MeV) are not greatly affected by the sawtooth, while low energy alphas ($E \sim 35$ keV) experience a strong outward radial transport. This is the ideal scenario to balance plasma self-heating of the core and helium ash removal.

This is not a fundamentally new idea, and the authors are careful not to claim that it is. They do a good job of citing and describing the previous work (with a few minor exceptions, see Comment 2 below). They claim that their model is the first to holistically capture the process previously only understood as an amalgamation of heuristic models. They also claim to be the first to explain some details of the process previously only known phenomenologically, like that high energy counter-passing particles are better confined than high energy co-passing particles.

One bit I must admit that I'm struggling with, is the claim that previous works understood that higher energy alphas could be better confined than lower energy if the "de-tuning" time was shorter than the crash in the case of the former and longer in the latter, but that they did not understand that this difference was due to drifts. What then did they ascribe the difference in de-tuning times to? To me, it seems self-evident. But perhaps I am reading with the sub-conscious benefit of a more modern understanding.

In general, the paper is very high quality: well-written, thorough, rigorous. It will make an excellent publication and can almost serve as a review, a one stop shop for future generations to understand the subject of alpha channeling.

I have a few minor comments and questions below, in addition to the question above, but see no reason that this paper should not be published.

Comments:

1. Reference for lines 81-84? It seems to be the premise upon which the paper is built and originators should be cited.

2. Other missing references to consider:

N. Fisch has published a lot on alpha channeling, which seems to be an undercurrent here. Although there is a clear differentiator here in that the channeling is not being done actively with RF, but passively but exploiting a naturally occurring relations event, consider N. Fisch. AIP Conference Proceedings 1689, 020001 (2015); doi: 10.1063/1.4936463

On the topic of physical consequences of differences between orbit and magnetic topology, consider G. Fiksel. Phys. Rev. Lett. 95, 125001

3. Line 172, “proof-of-principal study” seems to contradict line 137, “we complete the physical picture.” In my opinion, “proof-of-principal” is an undersell.

4. Why not use same color table and scale in 1b and 3e if they’re to be compared directly, as the caption for Figure 3 suggests?

5. Line 231: ... in such a way that the pressure peak of Fig. 1 (b) (not (c), as written)

6. In the caption for Figure 4, the midplane is referred to as $z_0 = 0.26$ m, elsewhere it is $z_0 = 0.27$ m (i.e., Line 252).

7. Line 245: this seems like a large range for τ_{crash} , 20x? Especially given that later in the paper you will look for parameters that give a detuning time as a function of energy that straddles this time. In fact, in Line 321, a much tighter range of $\tau_{\text{crash}} \sim 0.3$ ms is given. Consider quoting a smaller range or justifying the large range.

8. It is mentioned several times that $q \sim 1$ implies that magnetic drifts are significant. I understand that the drifts are responsible for the de-tuning, and furthermore, that having $q \sim 1$ implies that many transits are required for detuning, allowing for the situation in which high energy alphas de-tune and low energy alphas do not. Is that what is meant by the statement, e.g., “alpha particle resonances with

respect to the internal kink are sensitive to magnetic drifts" (lines 317-319)? Or are the authors saying that were q far from unity the fast ions would not drift relative to the magnetic surfaces?

9. Are finite Larmor radius effects not important? Is the Larmor radius not large enough to separate q and h ?

Reviewer #2 (Remarks to the Author):

This paper presents theoretical results predicting the existence of a parameter window where sawtooth oscillations can exhaust Helium ash with only small degradation in the confinement of MeV-class energetic particles. The theory extends previous analyses by including the effects of magnetic drifts, thereby accounting for the different response between co and counter-passing particles

The paper is well-written and the theoretical considerations it presents are extensively researched; the literature cited is well-developed but dominated by theory. Two weaknesses are (i) the absence of any comparison to experimental observations and (ii) the lack of any mention of the possible effects of plasma rotation, particularly zonal rotation driven by the loss of energetic particles. Other questions are whether the losses of energetic particles caused by precursor or postcursor oscillations during the sawtooth ramp could undermine the beneficial effects of the sawtooth, as may the seeding of neoclassical tearing modes and the poor electron thermal confinement that is observed in the sawtooth region for very flat q profiles. While the paper does not address these concerns, it concludes with a thoughtful discussion of other possible difficulties, showing that the authors are cognizant of many of the complexities of scenario development.

In summary, this paper presents an interesting idea for solving what is anticipated to be a serious problem for fusion reactors such as ITER. The paper stands out by its ambition, the resourcefulness of its authors and its implicit optimism. I recommend publication after the authors have had an opportunity to consider my comments.

Manuscript NCOMMS-22-01064: “**Energy-selective confinement of fusion-born alpha particles during internal relaxations in a tokamak plasma**” by Bierwage, Shinohara, Kazakov, Kiptily, Lauber, Nocente, Štancar, Sumida, Yagi, Garcia, Ide and JET Contributors

Response to referees

Naka / Japan, 21 April 2022

We are grateful to both referees for kindly evaluating our manuscript. Their constructive comments and encouragement were much appreciated.

Below we provide point-by-point answers. The reviewers’ comments are reproduced in blue italics. Note that the line numbers and reference numbers appearing in this answer letter refer to the original version of the paper. Changes made in response to the reviewers’ comments are printed in red in this letter. The corresponding changes in the revised version of the paper should be easy to find if one looks for the colored text in the enclosed PDF file.

Reviewer #1:

Summary and general comments:

A kinetic-MHD hybrid simulation is used to model alpha particle transport during a sawtooth relaxation event driven by an $m=1$, $n=1$ internal kink in a JET equilibrium. It is found that for certain parameters, namely q within a few percent of 1, the highest energy alphas ($E\sim 3.5$ MeV) are not greatly affected by the sawtooth, while low energy alphas ($E\sim 35$ keV) experience a strong outward radial transport. This is the ideal scenario to balance plasma self-heating of the core and helium ash removal.

This is not a fundamentally new idea, and the authors are careful not to claim that it is. They do a good job of citing and describing the previous work (with a few minor exceptions, see Comment 2 below). They claim that their model is the first to holistically capture the process previously only understood as an amalgamation of heuristic models. They also claim to be the first to explain some details of the process previously only known phenomenologically, like that high energy counter-passing particles are better confined than high energy co-passing particles.

One bit I must admit that I’m struggling with, is the claim that previous works understood that higher energy alphas could be better confined than lower energy if the “de-tuning” time was shorter than the crash in the case of the former and longer in the latter, but that they did not understand that this difference was due to drifts. What then did they ascribe the difference in de-tuning times to? To me, it seems self-evident. But perhaps I am reading with the sub-conscious benefit of a more modern understanding.

Thanks for pointing out the ambiguity. We think that we were able to identify (and eliminate) three possible causes of confusion.

First, our Introduction section did not separate cleanly between the review of prior work and our own findings. This has now been fixed (also in compliance with the editorial guidelines).

Second, we realized that we had not made clear that the words “decoupling” and “detuning” were used to refer to different things:

- “Detuning” refers to phase slippage due to rapid parallel streaming along the \mathbf{B} field.
- “Decoupling” referred to the combined effect of parallel streaming and magnetic drift.

Third, our discussion of the magnetic drift effect (causing **resonance shifts**) may not have been entirely clear and has been improved.

Please note that magnetic drift is not needed for the detuning via phase slippage. The detuning time is the time required for the particle orbit to undergo a 2π phase shift relative to the kink mode. That phase shift corresponds to one orange-blue cycle in Fig.6(d-f). Assuming that the toroidal phase of the kink mode remains more or less fixed, it takes

$$N(2\pi) \sim 1/|1-h| \sim 1/|1-q|$$

toroidal transits for the orbit to cover a full toroidal surface and, thus, to undergo a 2π phase shift. When the orbit helicity h (and q) is close to 1, this takes a long time. The motion of passing particles is similar to the apsidal precession of the moon around Earth shown in the picture on the right. [Taken from wikipedia.org.]

It is evident that the detuning time scale depends mainly on the parallel velocity component. The rest of the existing theory follows from comparing the detuning time with the sawtooth crash time. Within this scope, the magnetic drift contributes only a negligible quantitative correction to the detuning time $\tau_{2\pi}$. **The point that had been missed is that the magnetic drift affects the structure of the 1/1 resonance when $q \sim 1$.**

The magnetic drift had been taken into account in cases where its role is more obvious; e.g., for nonstandard orbits. One example is Ref. [37] by Kolesnichenko *et al.*, where large “potato” orbits were analyzed, which are important for heat loads on plasma facing components. In contrast, in their study of generic alpha particle transport during sawteeth, Kolesnichenko & Yakovenko clearly state in the first paragraph of Ref. [17] that they choose to employ a small-drift approximation. Hence, their theory is valid for passing alphas only in configurations where q is sufficiently far from 1. This makes the problem more easily tractable analytically, but excludes an interesting regime from the theory’s domain of validity.

The importance of the drift for the general population of alpha particles, and the interesting properties of $q \sim 1$, can be anticipated by inspecting the orbit helicity profiles $h(R | E, \mu)$ in our Fig.7. Figure 2 of the work by Fiksel et al. [PRL’05] mentioned by the reviewer is another good example. However, it seems that these instructive orbit helicity profiles h are not widely used, even though they are a straightforward extension of standard wave-particle resonance analysis. This may be another reason for why some implications have been overlooked, most notably the drift-induced resonance shift.

In the revised manuscript, we have limited the use of the word “decoupling” to the review of earlier work. When discussing the physical mechanisms that were identified in our simulations, the word decoupling is no longer used. Instead, we spell out the drift effect explicitly; e.g., as “drift-induced resonance shift”. We also reduced the use of the word “synergy”, which may sound profound but is not really illuminating.

Below are the concrete changes made. At the end of the Introduction section, we write:

“The important point to note then is this: When the magnitude of the parameter $|1 - q|$ drops to the level of a few percent, the resonance condition between alpha particles and the kink becomes sensitive to drifts associated with magnetic gradients (grad- B and curvature) even for passing alpha particles. This effect was not considered in the existing theory and it is the key insight underlying the present study. [...]

[...] The observations are explained in terms of a synergistic effect that emerges for a type of sawteeth where the magnetic field helicity remains close to unity ($q \sim 1$). An optimal crash time scale facilitates detuning of fast alphas from the internal kink, while helium ash remains phase-locked. However, detuning due to rapid parallel streaming along the \mathbf{B} field is effective only until the particles enter a resonant reconnection layer, which is why electron profiles are eventually flattened. It is then the magnetic-drift-induced difference between orbit topology and magnetic topology that allows the majority of fast alphas to sustain a peaked density profile throughout the reconnection process. This realizes the looked-for energy-selective confinement for all pitch angles.”

In the Results section “Physical mechanism” (previously called “Underlying synergism”), we write:

“Here, the physical picture is completed by including (ii) the magnetic-drift-induced shift of the resonances. The combination of factors (i)-(iv) in Fig. 6 facilitates a selective confinement of fast alphas and mixing of slow alphas.”

The captions of Fig. 6 and 7 were improved for clarity. The revised Results section also contains a few new paragraphs and a new Fig. 8. This figure 8 highlights our key finding that

- detuning via phase slippage, and
- sensitivity of resonances with respect to magnetic drift

are both necessary and that neither is sufficient on its own. The new figure should convey the message more clearly than the mere verbal discussion of the original manuscript. The text was also improved and expanded (page 7):

“[...] This detuning (phase slippage) combined with the magnetic-drift-induced resonance shift described earlier allows the density field of 3.5 MeV alphas in Fig. 4(f) to maintain a compact peak with only a minor and temporary helical distortion.

It can be verified that all ingredients are needed. For instance, the rapid motion of thermal electrons allows them to satisfy the detuning condition (iii) and the scale separation condition (iv) in Fig. 6. However, the lack of magnetic drifts causes electrons to stick closely to magnetic field lines and undergo strong mixing during sawtooth crashes that involve magnetic reconnection. This is confirmed in Fig. 8(a), which shows the redistribution of an artificial particle species with charge number $Z = 2$, reduced mass $M_{0.1} = 0.1 \times M_a$ and high speed $v = 13 \times 10^6 \text{ m s}^{-1}$. Its magnetic drift is as small as that of slow 35 keV alphas with mass M_a , while the transit frequency is as high as that of fast 3.5 MeV alphas. One can see that the density profile in Fig. 8(a) is flattened.

Figure 8(c) shows that the density profile is also flattened in the opposite limit, where heavy particles ($M_{10} = 10 \times M_a$, $Z = 2$) travel at low speed ($v = 1.3 \times 10^6 \text{ m s}^{-1}$). Here, the magnetic drift (and gyroradius) is as large as that of fast 3.5 MeV alphas with mass M_a , so that the density field in Fig. 8(d) is similarly blurred as in Fig. 4(f). Nevertheless, the density profile in Fig. 8(c) is subject to much stronger flattening than in Fig. 4(e). This must be due to the particle speed v being smaller by a factor 10 since all other parameters are identical.

The results in Fig. 8 thus show that the resonance shift due to the magnetic drift, which (for a given ν) is larger for particles with smaller charge-to-mass ratio Ze/M , is not sufficient but necessary for preventing profile flattening. Similarly, detuning due to rapid parallel streaming is necessary but not sufficient. This proves that the selective confinement of fast alphas in a wide range of pitch angles is realized only through the combination of the four factors summarized in Fig. 6.”

In general, the paper is very high quality: well-written, thorough, rigorous. It will make an excellent publication and can almost serve as a review, a one stop shop for future generations to understand the subject of alpha channeling.

Thank you for the encouragement. It is a great responsibility and we hope that this paper will serve this ambitious purpose.

I have a few minor comments and questions below, in addition to the question above, but see no reason that this paper should not be published.

(1) Reference for lines 81-84? It seems to be the premise upon which the paper is built and originators should be cited.

We had considered this when preparing the manuscript, but it turned out to be difficult to identify the originator of the idea. Suggestions are highly welcome.

For the time being, we decided to cite three works published in 1991:

- a technical paper by Redi & Cohen from PPPL [Fus.Tech. 20 (1991) 49], reporting numerical analyses including – among other things – helium transport analyzed by a code that contains a simple sawtooth model;

and two review-style papers,

- one by Rebut et al. from JET [PoF B 3 (1991) 2209], and
- one by Reiter et al., from FZJ [PPCF 33 (1991) 1579].

The references appear at the end of line 84.

However, the basic idea is probably older. It is often passed around by word-of-mouth and may have been spooking around ever since the Kadomtsev model of the sawtooth crash had been proposed, where it became evident that the phenomenon is a type of self-organization process. It seems to be a natural trait of human beings to try to put an apparent nuisance to good use. In that sense, a mere speculative proposal to use sawtooth crashes for helium ash removal does not appear to be a major step forward. The above-mentioned paper by Redi & Cohen seems to be one of the first to show concrete (numerical) data.

(2) Other missing references to consider:

N. Fisch has published a lot on alpha channeling, which seems to be an undercurrent here. Although there is a clear differentiator here in that the channeling is not being done actively with RF, but passively but exploiting a naturally occurring relations event, consider N. Fisch. AIP Conference Proceedings 1689, 020001 (2015); doi: 10.1063/1.4936463

Thanks for this suggestion. We have tentatively added the following sentence after line 89:

“In the case of so-called alpha channeling via radio-frequency (RF) waves [Fisch PRL’92, Fisch AIP-CP’15], it is envisioned that the alpha particles gradually diffuse outward while transferring their energy to thermal ions via damped plasma waves. A recent study shows, however, that the RF-cooled alphas may remain in the plasma core [White PoP’21a]. In contrast, resonant interactions with the internal kink and other low-frequency magnetohydrodynamic (MHD) modes can cause rapid ballistic transport before the alphas have transferred their energy to the bulk [White PoP’21b].”

We do however feel that it may be a little risky to mention **alpha channeling** in the present context. There are two reasons.

First, the alpha transport in Fisch’s theory is a diffusive one. It is a consequence of the (reasonable) assumption that the alpha density profile is peaked in the core (where they will be born) and decreases towards the edge (where they will be pumped out). The diffusive transport down a gradient is very different from the convective transport caused by a sawtooth crash. In particular, it seems that Fisch’s diffusive alpha channeling concept does not include any mechanism for channeling only slowed-down alphas radially outward while keeping fast alphas confined. Instead, the idea as we understand it is to perform stimulated cooling and radial transport simultaneously. In that sense, the connection with the present work is a little vague and we hope that mentioning alpha channeling here does not confuse the readers.

Second, the ability of RF waves to cause significant radial diffusion seems to be controversial. A study showing contrary evidence and explaining the reason for the discrepancy (namely, a certain term in the equations of motion) was recently published:

White et al., Phys. Plasmas 28, 012503 (2021); doi: 10.1063/5.0033497.

We are not in the position to settle that dispute and have cited both Fisch’s and White’s work.

In fact, the broader research community seems to associate Fisch’s proposal primarily with “alpha energy channeling”; namely, a way to bypass or complement the collisional heating channel via electrons. When I (A.B.) discussed the above work by White at a meeting last year, it caused some confusion, because I was speaking about the spatial transport aspect, while the audience seems to have anticipated to hear only about energy channeling.

The present manuscript bears no direct connection to alpha energy channeling.

On the topic of physical consequences of differences between orbit and magnetic topology, consider G. Fiksel. Phys. Rev. Lett. 95, 125001

Thank you for the suggestion. We agree that this paper should be cited here and have added the reference on line 338, in front of the old references [34, 35] (page 5).

(3) Line 172, “proof-of-principal study” seems to contradict line 137, “we complete the physical picture.” In my opinion, “proof-of-principal” is an undersell

Motivated by the reviewer’s comment, we removed the attribute “proof-of-principle” since it may indeed have a different nuance than intended.

(4) Why not use same color table and scale in 1b and 3e if they're to be compared directly, as the caption for Figure 3 suggests?

In an earlier draft, Fig.3(e) had the same color map and color scale as Fig.1(b). Here it is:

However, after I (A.B.) had presented these results at a meeting, I have been informed that panel (e) appeared featureless to colorblind colleagues. Although the “parula” colormap we used was meant to avoid such problems, it seems to be effective only when the full color scale is used as in Fig.1(b). Since the original Fig.3(e) showed only the orange-greenish part of the spectrum, that effect seems to have been lost. The current Fig.3(e) is meant to solve this problem.

(5) Line 231: ... in such a way that the pressure peak of Fig. 1 (b) (not (c), as written).

(6) In the caption for Figure 4, the midplane is referred to as $z_0 = 0.26$ m, elsewhere it is $z_0 = 0.27$ m (i.e., Line 252).

Thank you very much for spotting these typos. The midplane crosses the axis at $z_0 = 0.266$ m ≈ 0.27 m. We corrected the value in the captions of Figs. 3, 4, 7, and in the Methods section.

(7) Line 245: this seems like a large range for τ_{crash} , 20x? Especially given that later in the paper you will look for parameters that give a detuning time as a function of energy that straddles this time. In fact, in Line 321, a much tighter range of $\tau_{\text{crash}} \sim 0.3$ ms is given. Consider quoting a smaller range or justifying the large range.

The paragraph ending at line 321 describes why good confinement of fast alphas is seen in our simulation. The attribute “typical” in that sentence was a slip of the pen. Thanks for pointing this out. The corrected sentence reads

“[...] shorter than (iv) the ~~typical~~ sawtooth crash time $\tau_{\text{crash}} \sim 0.3$ ms in Fig.3. [...]”

A similar modification was made on lines 413–414:

“Sawteeth with crash times τ_{crash} of a few 100 μ s as in Fig. 3 are then shorter than [...]”

(8) It is mentioned several times that $q \sim 1$ implies that magnetic drifts are significant. I understand that the drifts are responsible for the de-tuning, and furthermore, that having $q \sim h \sim 1$ implies that many transits are required for detuning, allowing for the situation in which high energy alphas de-tune and low energy alphas do not. Is that what is meant by the statement, e.g., “alpha particle resonances with respect to the internal kink are sensitive to magnetic drifts” (lines 317-319)? Or are the authors saying that were q far from unity the fast ions would not drift relative to the magnetic surfaces?

The answer to the last question is: No. Fast ions are subject to grad- B drifts, but when q is far from unity, the relative difference $|h - q| / |1 - q|$ becomes too small for these drifts to influence the existence and location of $h = 1$ resonances. Only when $q \sim 1$ are the drifts able to eliminate $h = 1$ resonances (for counter-passing fast alphas) or reduce the reconnected domain in orbit topology (for co-passing fast alphas). This can be inferred by comparing the helicity profiles $h(X)$ in Fig.7 of the main article with those in the Supplementary Figs.S10(n), S11(n).

We suspect that the reviewer’s statement saying that “drifts are responsible for the de-tuning” refers to what we had called “decoupling” rather than “detuning”. We realized that the distinction had not been made clear in the original text and hope that our reply to the reviewer’s “Summary and general comments” at the beginning of this letter has clarified the matter. Let us repeat the main points:

Only for trapped particles the detuning time is determined by magnetic drifts – namely, the precessional drift around the torus. The confinement of trapped particles is described well by the existing theory. Trapped particles are present in our simulations and their behavior in Fig.5 is consistent with the existing theory. Our analysis **focuses on co- & counter-passing alphas**, which dominate in the central plasma, and for which the existing theory was incomplete.

In the case of passing particles, the condition “ $q \sim h \sim 1$ ” has two consequences as is implied by the two arrows emanating from box (i) in Fig.6:

- (ii) The existence and location of resonances is sensitive to magnetic drifts.
- (iii) The detuning times of fast and slow alphas differ by an order of magnitude.
(... so it is possible to (iv) have sawtooth crash times sandwiched between them.)

For passing ions, the detuning time $\tau_{2\pi}$ is primarily set by parallel streaming. The transverse drifts contribute only a minor correction to the value of $\tau_{2\pi}$. Meanwhile, item (ii) is one of the most important points that we would like to convey in this paper, so it appears in the abstract: **“Besides causing asymmetry between co- and counter-going particle populations, magnetic drifts determine the size of the confinement window by dictating where and how much reconnection occurs in particle orbit topology.”**

Another important point is that large drifts are necessary but not sufficient. We have demonstrated this by performing a numerical experiment where the particle mass was varied. In the original manuscript, this was mentioned only verbally on lines 435–443 and 457–460. In the revised manuscript, we have added those results in visual form with a new Figure 8.

(9) Are finite Larmor radius effects not important? Is the Larmor radius not large enough to separate q and h ?

We did not touch this topic in the main part of the paper, but there was a comment in the Methods section saying that **finite Larmor radius (FLR) effects are not important here**:

“Our simulations were run with 4-point gyroaveraging around a particle’s guiding center [Bierwage NF’16c]. Very similar results can be obtained in the zero-Larmor-radius limit, even for 3.5 MeV alpha particles.”

We have tested this by running the simulations with and without gyroaveraging. A related note was also made on lines 207–213 in the original Supplementary Material.

In order to make this information easier to find, the revised Supplementary Information file now contains a table of contents, and there is a dedicated Section “6. Gyroaveraging”. In the Methods section of the main paper, we created a separate subsection “Gyroaveraging”, where the above paragraph is followed by further explanation:

“The reason for why the finite Larmor radius (FLR) has no notable effect here is that the fast alphas in our simulations were already well-confined without gyroaveraging, while the slow alphas that underwent strong mixing have small gyroradii.

It is also worth noting that gyroaveraging and magnetic drifts work in different ways. Gyroaveraging occurs effectively instantaneously around a given guiding center position, so that this aspect of FLR has merely a local smoothing effect. It tends to reduce the effective magnitude of the electric field and, consequently, the electric drift velocity and energy transfer, but it hardly affects the resonances as such. In contrast, the magnetic drift displaces the entire guiding center orbit contour relative to the magnetic flux contours, so its effect is global. In combination with the mirror force, the magnetic drift can shift or even eliminate resonances (Fig. 7). The commonly held notion of a ‘drift-orbit-averaging effect’ can hence be misleading. In principle, the magnetic drift may even enhance the resonant coupling between an MHD mode and fast particles, because it can localize the interaction domain poloidally [Zhang NF’15] and, thus, maximize phase locking (minimize phase shift), albeit at the expense of reducing the interaction time.

We expect that FLR effects such as gyroaveraging will be more important when the population of alpha particles is large enough to influence MHD modes, which was not the case here. The large gyroradii of fast alphas are expected to reduce the efficiency of resonant coupling with MHD modes [Bierwage NF’16c].”

A point worth noting in this context is that the orbit helicity profile h is not determined by the magnetic drift alone, but also by the mirror force as was explained on lines 337–359 of the paper. Without the mirror force, co- and counter-passing particles would have the same helicity profile h . Given the grad- B -shifted orbits, it is then the modulation of v_{\parallel} due to the mirror force that raises h for counter-passing alphas and decreases h for co-passing alphas, because the particles spend more time on the high-field side than on the low-field side. Orbits near the stagnation points (= drift-kinetic counterparts of the magnetic axis) are the extreme example: they only see the local magnetic field line pitch at a certain poloidal angle: $\theta = 0$ for co-passing, and $\theta = \pi$ for counter-passing.

We do not recall having seen this fact mentioned in other works – not even in those that utilize orbit helicity profiles. Of course, both effects – grad- B drift and mirror force – always coexist in a toroidal plasma. However, we think that it helps to consider their respective roles.

Reviewer #2:

Summary and general comments:

This paper presents theoretical results predicting the existence of a parameter window where sawtooth oscillations can exhaust Helium ash with only small degradation in the confinement of MeV-class energetic particles. The theory extends previous analyses by including the effects of magnetic drifts, thereby accounting for the different response between co and counter-passing particles

The paper is well-written and the theoretical considerations it presents are extensively researched; the literature cited is well-developed but dominated by theory. Two weaknesses are (i) the absence of any comparison to experimental observations

Experimentally measured sawtooth crash times and sawtooth inversion radii play a part in our study, so comparisons with experimental observations are not entirely absent. We understand however that the reviewer’s critique refers to the absence of an **experimental test** for our prediction of selective confinement of fast alphas and mixing of ash for sawteeth with $q \sim 1$.

Such an experimental test has not been performed yet because it depends on breakthroughs in plasma control and diagnostics. The test would require a systematic experimental study of fast ion transport during sawteeth in a specific parameter regime with central safety factor $q \sim 1$. Among various challenges, two stand out:

- Experimental operation in a regime with $q \sim 1$ requires advances in sawtooth control that, in turn, depend on a better understanding of sawtooth physics (lines 504–512).
- Experimental diagnostics that allow sufficiently accurate measurements of the safety factor q in the core remain to be developed.

These items are prerequisites for a conclusive test of our predictions. Ways to overcome these hurdles would constitute major breakthroughs on their own. The first item was already mentioned on lines 504–512. We have modified that text somewhat and include the mention of the need for better diagnostics (Discussion section, now on page 8):

“In order to **experimentally validate and** utilize the **sawtooth-mediated** energy-selective mixing and confinement in a reactor, it is thus necessary to ensure that only certain types of sawteeth occur, namely those for which the field helicity remains close to unity ($q \sim 1$). ~~Provided that this is practically possible, it~~ This requires **more accurate q profile diagnostics**, a better understanding of sawtooth **physics**, and **more** precise plasma control schemes, where one controls not only the sawtooth period but also the form of the crash. **The benefits of advancing these capabilities go far beyond the subject of the present work: Besides enabling alpha particle density control, another important advantage of operating reliably in a regime with benign sawteeth and $q \sim 1$ is that it minimizes the forced formation of magnetic islands in the outer core and peripheral plasma (see Supplementary Fig. S4), which is important for maximizing performance and avoiding plasma disruptions.”**

Some supporting experimental evidence does exist, most notably the DIII-D results reported by Muscatello *et al.* [18] that we mention in the Introduction section. A similarly systematic study will have to be carried out for fast alpha particles in a sawtooth plasma with $q \sim 1$. Ideally, those tests should be done in D-T plasmas, possible only at ITER, and perhaps again at JET. One purpose of the present work is to motivate and guide those efforts.

Opportunities to operate with fusion born alpha particles in significant quantities are rare and rely also on financial and political support. We believe that our work can also contribute to securing that kind of support.

In principle, the behavior of alpha particles may be partially emulated using deuterons, which have the same charge-mass ratio. In that case, one needs to find ways to produce multi-MeV-class deuterons and scan the full range of pitch angles to test our predictions. And, in order to observe energy-selective transport, one needs energy-selective transport diagnostics for ions. The fact that ion density profiles are usually inferred indirectly highlights that this is a tough problem, which is further complicated by the short time scale of a sawtooth crash. The lack of simultaneously good spatial and good temporal resolution is a common shortcoming of prior experiments, where fast ion confinement during sawtooth crashes was studied.

However, we think that the problem of monitoring (or emulating) alpha particle transport is secondary to the problem of ensuring that the plasma is in the correct state ($q \sim 1$) as discussed above. Precise control and accurate measurements of the q profile seem to be absolutely essential for a conclusive experimental test.

and (ii) the lack of any mention of the possible effects of plasma rotation, particularly zonal rotation driven by the loss of energetic particles.

Although we have not spent many words on this, **rotation** was mentioned on lines 516–519:

“The same counts for the influence of sawtooth-induced alpha particle transport on the background plasma from the viewpoints of heating, current drive and plasma rotation.”

The rest of that paragraph discussed the generation of sheared toroidal torque due to an imbalance between co- and counter-going particles, since that imbalance (caused by magnetic drift) is one of the topics of the present work.

If we understand correctly, Reviewer #2 refers to one particular source of rotation: **ambipolar transport**; namely, plasma rotation caused by ion losses exceeding electron losses. There are two reasons for why we do not mention this effect in the present paper:

- **Fast ion losses are negligibly small** (even for 3.5 MeV alphas) in the central core of a large tokamak subject to benign sawtoothing as is considered here. The subject of loss-induced zonal flows seems to be more relevant for the plasma closer to the edge.
- The subject of ambipolar transport and the **associated problem of electron confinement is still poorly understood and a matter of ongoing research.**

In a reactor, instead of considering losses, one may speculate that charge imbalance may arise from D-T fusion reactions, as they act as a sink for thermal D and T ions (small radial drifts) and a source for 3.5 MeV alphas (large radial drifts). However, this speculation assumes that magnetic drifts alone (without actual losses) can contribute to the formation of a significant radial electric field and associated zonal flows, which might not be the case. Since the larger drifts of fast ions make them pass over a wider range of magnetic surfaces, electrons can respond to some extent by streaming along field lines, while staying on their flux surfaces.

Although the parallel electron response may not suffice to entirely suppress the build-up of a radial electric field, one should also consider the possibility of electrons traveling across magnetic surfaces. Ubiquitous magnetic flutter may give electrons a certain amount of freedom to travel radially, allowing them to short out remaining charge imbalance caused by the radial displacement of ions. The above-mentioned “magnetic flutter” includes nonideal high-beta equilibrium physics and the magnetic components of drift-Alfvén modes, which can have mixed parity (tearing and interchange). This includes fast-ion-driven shear Alfvén modes in high-beta plasmas as well as thermal-gradient-driven electromagnetic turbulence. Phenomena associated with the buzzword “micro-tearing” also fall into this category. The reverse effect – enhanced electron confinement – is apparently connected with transport barrier formation. All these are relatively immature subjects that await a deeper theoretical understanding and predictive capability.

Electron confinement (and consequences for ambipolarity) is currently a topic of active research. Global nonlinear electromagnetic simulations of electron transport are now possible, but seem to be still in their infancy, especially with respect to the formation of a “self-consistent kinetic equilibrium” (if such a thing exists on the relevant scales) including intrinsic rotation. Computational physicists developing and testing these codes seem to be still struggling with various technical (and perhaps conceptual) problems. There are ongoing efforts to develop tractable models;

e.g., Chen et al., PPCF 61 (2019) 035004, <https://doi.org/10.1088/1361-6587/aaf42d>.

Therefore, we hesitate to raise the topic of ambipolar transport in the present paper. We should wait for a better understanding of electron confinement in toroidal plasmas.

Other questions are whether the losses of energetic particles caused by precursor or postcursor oscillations during the sawtooth ramp could undermine the beneficial effects of the sawtooth, as may the seeding of neoclassical tearing modes and the poor electron thermal confinement that is observed in the sawtooth region for very flat q profiles. While the paper does not address these concerns, it concludes with a thoughtful discussion of other possible difficulties, showing that the authors are cognizant of many of the complexities of scenario development.

Motivated by this comment, we felt that these points deserve more emphasis and added the following paragraph to the Discussion section (page 8):

“Other factors requiring consideration are the bulk plasma confinement in regions where $q \sim 1$, plasma rotation, and the role of pre- and post-cursor oscillations. While important for sawtooth physics, it must be noted that the helical distortions observed as pre- and postcursor oscillations in a rotating plasma do not necessarily affect overall confinement. On the collisionless time scales where adiabatic invariants of guiding center motion are valid, confinement can be broken only by resonant interactions [Chen RMP’16, Chen PST’19] via convective phase space instabilities [Berk PLA’97, Zonca NF’05, Zonca NJP’15], resonance overlaps [Berk NF’95], or reconnection phenomena (as in the present work).”

Concerning **neoclassical tearing modes (NTM)**: The avoidance of NTMs and their side effects (ranging from reduced plasma performance to major disruptions) is important and perfectly in line with our targeting of **benign sawtooth crashes** with $q \sim 1$. Although we did not mention NTMs explicitly, our Supplementary Material mentioned disruptions, which is one of the most serious threats of NTMs and other reconnecting modes in the outer core and plasma periphery (e.g., resistive wall modes). Specifically, Supplementary Fig. S4 shows that our benign sawtooth crash gives rise to relatively small secondary islands – here, through resistive reconnection. This was accompanied by the following discussion:

“Conversely, in the full domain, the kink can cause distortions of other low-rational resonant surfaces, such as $q = 3/2, 2/1, 3/1, \dots$, and lead to the formation of magnetic islands (or enlarge existing ones) through driven magnetic reconnection. This is an actual problem in real plasmas, sometimes leading to major disruptions that terminate the discharge. For this reason, giant sawtooth crashes should be avoided. This was, in fact, another motivation for us to focus on cases with $q \sim 1$. Nevertheless, even the benign sawtooth crash in our simulations with $q_0 = 0.98$ does cause magnetic reconnection at other resonant surfaces as shown in S.Fig. 4. These islands are however sufficiently small and sufficiently far apart to avoid overlaps, the spread of chaos and associated loss of confinement.”

A similar note has now been added in the Discussion section as mentioned earlier on page 9 of this letter:

“[...] Besides enabling alpha particle density control, another important advantage of operating reliably in a regime with benign sawteeth where $q \sim 1$ is that it minimizes the forced formation of magnetic islands in the outer core and peripheral plasma (see Supplementary Fig. S4), which is important for maximizing performance and avoiding plasma disruptions.”

Questions regarding the quality of **electron confinement in plasmas with $q \sim 1$** are related to our above reply to the question about plasma rotation caused by ambipolar transport.

Enhanced electron mobility in the radial direction may suppress the build-up of radial electric fields. However, as discussed earlier in this letter, we need to wait until scientists have refined their numerical tools and gained a better understanding of electron confinement before we can say anything definitive. There seem to be other open questions concerning $q \sim 1$, which is why our paper concluded with a generic statement about that matter (page 8):

“[...] field helicity profile $q(r)$ close to unity – lies precisely in the regime envisioned for ITER long-pulse operation. At the same time, this regime **with $q \sim 1$** still exhibits unsolved mysteries and, thus, opportunities [*... followed by some references*].”

Finally, we come to the topic of **pre- and postcursor oscillations**, which, in our opinion, is interesting for sawtooth control and their theoretical understanding, but not so much from the viewpoint of confinement as stated in the new paragraph of the Discussion section (cited above on page 11 of this letter). Let us try to explain the underlying rationale.

Observations of such pre- and postcursor oscillations imply that there exists a helical distortion inside the plasma (while the plasma is rotating past the probe). Although such a helical distortion breaks the axisymmetry of the tokamak plasma, it does not automatically imply a significant reduction of confinement (otherwise the stellarator program would supposedly have ended long ago). Overall confinement can be affected by nonresonant symmetry breaking only on longer time scales, since invariants of guiding center motion are not exact (only adiabatic), and since collisions will contribute.

The fact that electrons in experiments remain well-confined throughout the pre-cursor oscillations until the crash constitutes evidence showing that bulk confinement does not deteriorate significantly in that helically distorted state. It is reasonable to assume that the same is true for the majority of fast ions. In fact, this is precisely the reason why the confinement of our counter-passing alphas is not affected even by the sawtooth crash in the simulations: there is no resonance (and, thus, no reconnection) in counter-passing orbit topology. The helical distortions of the orbit surfaces as seen in our Fig. 6(b) is largely reversible on this time scale. This is explained in the paper on the basis of Fig. 7.

The only exception are particles traveling on orbits that happen to resonate with the kinked core or whose orbits are fragile for other reasons. In particular, this is the case for so-called “potato” orbits near the trapped-passing boundary, whose losses may be detected by external probes while the plasma core is helically distorted. While these “enhanced prompt losses” are important for diagnostics purposes, and possibly for considerations of wall safety and impurity control, the number of lost particles is small compared to the number of particles that remain well confined. Therefore, we assume that these losses are not a major concern within the scope of the present study.

Significant losses on short time scales (millisecond) occur only when there is a change in orbit topology, such as reconnection or resonance overlap (leading to chaos). Kink modes that resonate with fast particles are known as “fishbone oscillations”. These low-frequency modes tend to resonate with trapped fast ions (and even electrons) via their precessional drift. Fishbone oscillations cause large losses when those resonances become highly populated by beams and ICRH. They may be of lower concern for isotropically distributed alpha particles, but this must of course be examined case-by-case for the entire zoo of Alfvén modes, as already stated in the original text (lines 513–516; now in the lower half of page 8).

In summary, this paper presents an interesting idea for solving what is anticipated to be a serious problem for fusion reactors such as ITER. The paper stands out by its ambition, the resourcefulness of its authors and its implicit optimism. I recommend publication after the authors have had an opportunity to consider my comments

The reviewer raised important points and we hope that our replies are satisfactory. While ITER is in the making, we will encourage colleagues to develop the methods needed to test at least some of our theoretical predictions experimentally on existing machines and we will, of course, participate in such work ourselves.

* * *

List of changes, highlighted in color in the revised manuscript

Red: Changes made in response to referee comments.

Blue: Additional minor changes in notation and wording to improve clarity and to comply with editorial guidelines. Corrected typos and grammar. Changes worth noting:

- Revised all figure captions.
- Replaced “last closed flux surface” with “edge” in Table I.
- Changed symbol for charge number: “Q” → “Z”.
- Removed an unnecessary and speculative remark about the influence of E_{\parallel} in fast ions, which is most likely negligibly small (lines 360–363 of the original manuscript):
“These differences between the field and orbit helicities have two consequences. First, the parallel electric field E_{\parallel} in the magnetic reconnection layer near $q = 1$ has less influence on faster ions.”

Changes in References:

- Updated [36].
- Added [9–13, 39, 51– 57] in the main text, [70] in the Methods section, and [71] in the Data Availability section.

Changes in Supplementary Information: Changes are highlighted in blue color.

- Added a short abstract and a table of contents.
- Divided old Section 5 into new Sections 6 and 7, and moved the remainder to the main article.
- Correction in Supplementary Fig.S7 and related discussion: “ $f_{(II)} - f_{(I)}$ ” → “ $f_{\text{after}} - f_{\text{before}}$ ”.

REVIEWERS' COMMENTS

Reviewer #1 (Remarks to the Author):

Thank you to the authors for their thorough response and thoughtful edits. I now appreciate the difference between detuning and decoupling.

I believe the manuscript is now suitable for publication in nature Communications.

Reviewer #2 (Remarks to the Author):

The authors have addressed all my concerns. Together with the changes suggested by the other referee, the paper is significantly improved. It is a pleasure to recommend publication.

Manuscript NCOMMS-22-01064A: “**Energy-selective confinement of fusion-born alpha particles during internal relaxations in a tokamak plasma**” by Bierwage, Shinohara, Kazakov, Kiptily, Lauber, Nocente, Štancar, Sumida, Yagi, Garcia, Ide and JET Contributors

Response to referees

Naka / Japan, 9 June 2022

We wish to thank the referees one more time for their effort and support.

Reviewer #1:

Thank you to the authors for their thorough response and thoughtful edits. I now appreciate the difference between detuning and decoupling.

I believe the manuscript is now suitable for publication in nature Communications.

Reviewer #2:

The authors have addressed all my concerns. Together with the changes suggested by the other referee, the paper is significantly improved. It is a pleasure to recommend publication.

* * *

List of changes, highlighted in color in the revised manuscript

Red: Minor changes in wording and small additions made for clarity.

Changes in Figures

- Replaced “(a.u.)” with “(arb.units)” in Figs. 4, 8.
- Enhanced quality of plots in Fig. 6.

Changes in References:

- Included titles and full list of authors.
- Corrected [36].
- Added [40].

Changes in Supplementary Information: Changes are highlighted in blue color.

- Replaced “(a.u.)” with “(arb.units)” in Figs. S3c, S13, S14.
- Added a section containing a list of “JET Contributors”.